# RapidRat: Development, validation and application of a genotyping-by-sequencing panel for rapid biosecurity and invasive species management

**Bryson M. F. Sjodin[1], Robyn L. Irvine[2], Michael A. Russello**[1] *

**1** Department of Biology, University of British Columbia, Kelowna, British Columbia, Canada, **2** Gwaii Haanas National Park Reserve, National Marine Conservation Area Reserve and Haida Heritage Site, Parks Canada, Skidegate, British Columbia, Canada

* michael.russello@ubc.ca

**Data Availability Statement:** SNP genotypic data are deposited in DRYAD (https://doi.org/10.5061/dryad.fxpnvx0p4). All Illumina raw reads are

## Abstract

Invasive alien species (IAS) are among the main causes of global biodiversity loss. Invasive brown (*Rattus norvegicus*) and black (*R. rattus*) rats, in particular, are leading drivers of extinction on islands, especially in the case of seabirds where >50% of all extinctions have been attributed to rat predation. Eradication is the primary form of invasive rat management, yet this strategy has resulted in a ~10–38% failure rate on islands globally. Genetic tools can help inform IAS management, but such applications to date have been largely reactive, time-consuming, and costly. Here, we developed a Genotyping-in-Thousands by sequencing (GT-seq) panel for rapid species identification and population assignment of invasive brown and black rats (RapidRat) in Haida Gwaii, an archipelago comprising ~150 islands off the central coast of British Columbia, Canada. We constructed an optimized panel of 443 single nucleotide polymorphisms (SNPs) using previously generated double-digest restriction-site associated DNA (ddRAD) genotypic data (27,686 SNPs) from brown (*n* = 295) and black rats (*n* = 241) sampled throughout Haida Gwaii. The informativeness of this panel for identifying individuals to species and island of origin was validated relative to the ddRAD results; in all comparisons, admixture coefficients and population assignments estimated using RapidRat were consistent. To demonstrate application, 20 individuals from novel invasions of three islands (Agglomerate, Hotspring, Ramsay) were genotyped using RapidRat, all of which were confidently assigned (>98.5% probability) to Faraday and Murchison Islands as putative source populations. These results indicated that a previous eradication on Hotspring Island was conducted at an inappropriate geographic scale; future management should expand the eradication unit to include neighboring islands to prevent re-invasion. Overall, we demonstrated that RapidRat is an effective tool for managing invasive rat populations in Haida Gwaii and provided a clear framework for GT-seq panel development for informing biodiversity conservation in other systems.

available from the NCBI sequence read archive (BioProject ID: PRJNA633641).

**Funding:** This work was funded by Parks Canada (https://www.pc.gc.ca) agreement # GC-853 and Natural Sciences and Engineering Research Council of Canada (NSERC; https://www.nserc-crsng.gc.ca) Discovery grant # RGPIN-2014-04736 to MAR. The funder assisted in sample collection and reviewed drafts of the manuscript.

**Competing interests:** We note that author Michael Russello is an Academic editor for PLOS ONE; this does not alter our adherence to PLOS ONE policies on sharing data and materials.

## Introduction

Human modification of the environment has greatly accelerated the rate of global biodiversity loss in recent decades [1,2]. A leading driver of these losses is the introduction of non-native species to new environments. Once established, these invasive alien species (IAS) can alter ecosystem dynamics, especially through predation, competition, and disease transmission, resulting in rapid declines to endemic species [3,4]. IAS effects on biodiversity are profound; 58% of extinctions listed in the International Union for Conservation of Nature (IUCN) Red List name IAS as main contributors [5]. IAS also persist as economically important pests, costing Canada between $16.6 and $34.5 billion dollars annually [6], with approximately $20 billion of that in the Forestry sector alone [7]. Likewise, the United States spends >US$220 billion annually in prevention efforts, damages, and habitat restoration [8]. These negative effects are amplified within island systems, evidenced by IAS being the leading cause of the majority (86%) of recent island extinctions [3]. Moreover, insular systems support disproportionately high levels of endemism relative to mainland areas [9], punctuating the need for immediate action towards improving IAS management on islands.

Invasive mammals are the main cause of animal extinctions on islands and constitute one of the most significant threats to insular biodiversity [10]. Brown rats (*Rattus norvegicus*), black rats (*R. rattus*), and Pacific rats (*R. exulans*), in particular, make up a triad of the most invasive mammals on the planet, found on every inhabited continent and >80% of all oceanic islands [11]. To date, invasive rats have caused >50 species extinctions and are a leading cause of extinction for insular species [12–14]. Island seabirds, in particular, are adversely affected by rat invasions, as these species are poorly adapted to terrestrial predators due to their natural absence in many island systems [11,15,16]; in fact, approximately 60% of all seabird extinctions are linked to predation by invasive rats, 90% of these occurring on islands [14]. As part of seabird recovery efforts, management agencies often use whole-island eradications to manage invasive rat populations [14,17]. Yet, rodent eradication attempts on islands throughout the world have resulted in a ~10–38% failure rate [14,18]. In many cases, failure was due to knowledge gaps of key population parameters, including IAS population size, home range size, and dispersal capacity [14,18].

Genetic and genomic tools provide promising opportunities to inform vertebrate IAS management on islands. For example, genetic evidence was a large factor in the success of the largest rodent eradication in history on South Georgia Island (~390,000 ha) in the British Overseas Territory of the south Atlantic [19,20], providing critical information on population structure and demographic history that allowed for effective planning of a multi-stage eradication. In other cases, genetic and genomic tools can be used to explicitly evaluate the underlying cause of eradication failure, summarized in two prevailing hypotheses [14,21,22]. The survivor hypothesis suggests that the eradication effort failed to remove all individuals, leading to these individuals repopulating the island. On the other hand, the re-invader hypothesis posits that the eradication was successful in the complete removal of rats, but that the island was re-colonized from some connected source population. Genetic evidence was used to test these hypotheses following an unsuccessful rat eradication on Pearl Island in New Zealand; genetic analyses of population structure and individual assignment among pre- and post- eradication samples revealed evidence for re-colonization from a neighboring source population on Stewart Island (*i.e.*, re-invader hypothesis), suggesting that a larger target area may be required to avoid eradication failure in this system in the future [23]. Although useful for informing IAS management, genetic analyses to date have been largely reactive (*e.g.*, evaluating causes of eradication failure), time-consuming, and costly. Consequently, there is a need to harness leading-edge genomic technologies for providing genetic tools to aid managers in the

decision-making process before, during and after IAS management to help improve efficiency, eradication durability, and cost-effectiveness.

Here, we developed a Genotyping-in-Thousands by sequencing (GT-seq) panel for rapid species identification and population assignment of invasive brown and black rats (RapidRat) in Haida Gwaii, an archipelago off the central coast of British Columbia (BC), Canada. GT-seq is a targeted, multiplexed amplicon sequencing approach that can simultaneously geno-type sample sizes ranging from a single individual up to thousands of individuals at hundreds of single nucleotide polymorphisms (SNPs) [24]. We validated the RapidRat panel by com-paring estimated population structure and assignment accuracy with results from a previously published double digest restriction site-associated DNA sequencing (ddRAD) analysis (n = 27,686 SNPs) of the same samples within this system [25]. Lastly, we deployed RapidRat to identify the source of recent, novel brown rat invasions on Agglomerate, Hotspring, and Ramsay Islands.

## Methods

### Study system

Haida Gwaii (X̲aayda Gwaay in Haida) is an isolated archipelago consisting of approximately 150 smaller and two larger islands located ~80 km off the central coast of British Columbia (BC), Canada. This system contains more unique endemic sub-species than any other equal sized area in Canada [26] and is host to large numbers of seabirds and globally critical breeding habitat [27–32]. Both brown and black rats have invaded Haida Gwaii, likely as stowaways on European ships using the archipelago as a fishing and whaling port in the 1700's to early 1900s [29,30,33,34]. Since their introduction, invasive rats have adversely affected native seabird pop-ulations, resulting in the extirpation of several large multi-species colonies and population declines in at least half of the species present [29,31]. To meet seabird recovery goals and to protect ecological and cultural integrity [35], Parks Canada and other agencies started manag-ing invasive rats in the 1990s primarily through whole-island eradications. Since 1997, invasive rats have been successfully eradicated from five islands throughout the archipelago (Fig 1); however, brown rats were re-detected on the Bischof Islands following two eradication attempts in 2003 and 2011, and brown rats were detected on Faraday and Murchison Islands four years after a successful black rat eradication in 2013 [29,30,36,37]. Brown rats have now also spread to Hotspring, Ramsay, and Agglomerate Islands, all of which have been historically rat-free (C. Bergman, pers. comm.). The origins of these invasions are unknown, though both Agglomerate and Hotspring Islands are situated ~1 km from the nearest brown rat population on Murchison Island (Fig 1), while the newly established population on Hotspring Island could have invaded Ramsay Island (900-1000m away); these spans are within the hypothesized swimming distance for brown rats in temperate waters [25,38,39]. Additionally, rats could have potentially dispersed from island to island by floating on debris (i.e., "rafting") during high tide [40], and/or by commensal spread on ships.

We previously characterized population connectivity among brown and black rat popula-tions in Haida Gwaii using ddRAD sequencing (n = 27,686 SNPs) across the archipelago [25]. We found discrete genetic units largely corresponding to island populations, and we characterized within-island dynamics on the larger islands of Graham, Lyell, and Kunghit. We also assigned individuals from recent invasions on Bischof, Faraday, and Murchison Islands as invaders from neighbouring Lyell Island, demonstrating the utility of genetic analysis in this system, and providing an excellent source of reference data for the current study [25].

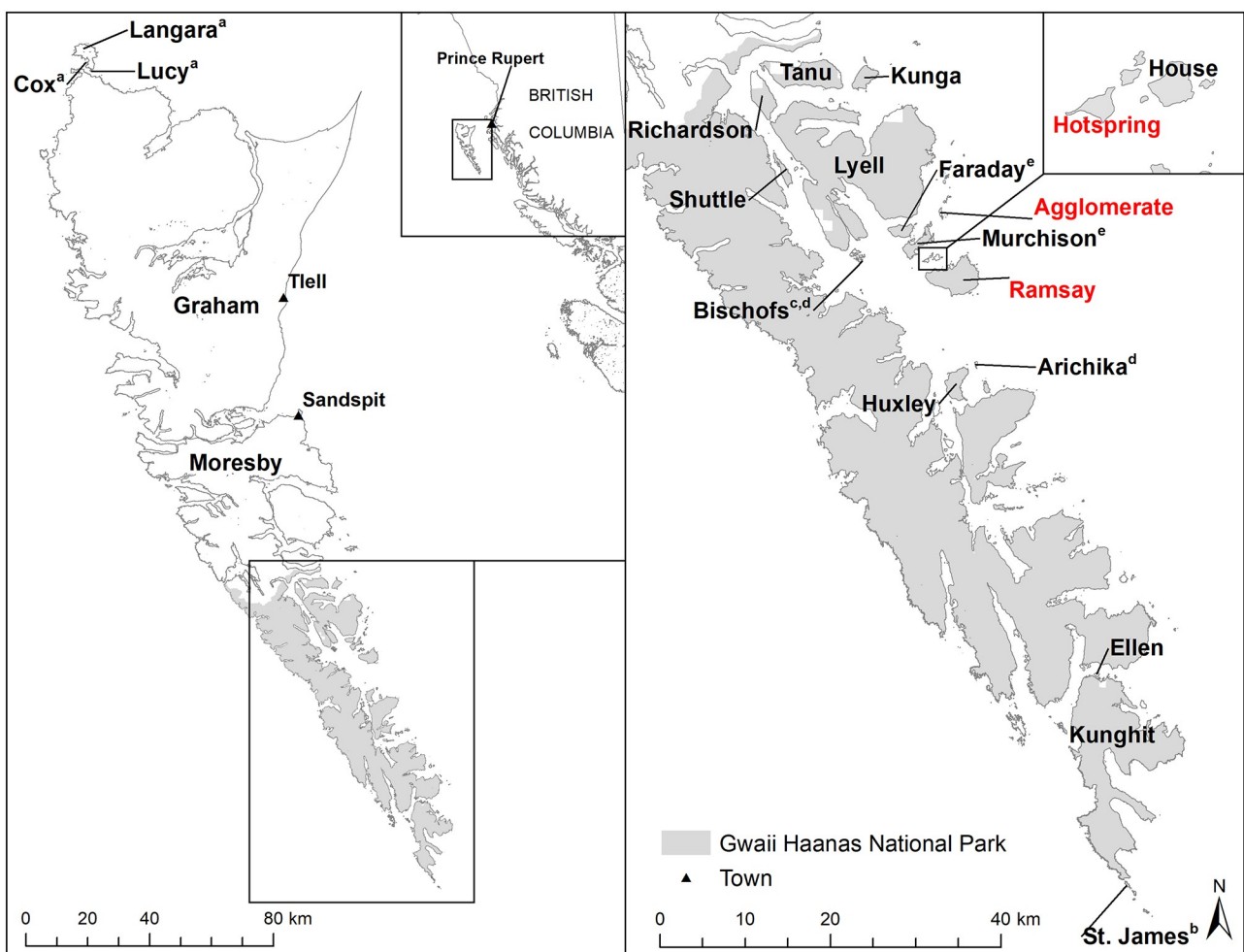

**Fig 1. Sampled islands and other locations within Haida Gwaii, British Columbia.** Island names with superscripts indicate past invasive rat (*Rattus* spp.) eradications. Eradications were performed in: (a) 1997; (b) 1998; (c) 2003; (d) 2011; and (e) 2013. Shaded are indicates the Gwaii Haanas National Park Reserve, National Marine Conservation Area Reserve and Haida Heritage site. Samples from islands written in red were used for GT-seq panel validation. The map was generated using data licensed under the Open Government Licence–British Columbia (https://www2.gov.bc.ca/gov/content/data/open-data/open-government-licence-bc).

## Study design

To identify candidate loci for our GT-seq panel, we used ddRAD data previously generated from brown rats ($n$ = 295) and black rats ($n$ = 241) collected throughout the Haida Gwaii archipelago [25]. Of these samples, brown rat ($n$ = 59) and black rat ($n$ = 37) DNA extractions (used in [25]) were re-sequenced and genotyped using GT-seq to optimize panel loci (see below). In addition to these samples, we sequenced and genotyped brown rats using GT-seq from novel invasions on Agglomerate Island ($n$ = 18), Ramsay Island ($n$ = 1), and Hotspring Island ($n$ = 1). See Table 1 for detailed sample distribution. All sample collection was performed under Parks Canada Agency Animal Care Committee protocol GHNPR11-5. Whole genomic DNA (gDNA) was extracted from 10–20 mg of dried ear tissue using the Qiagen DNeasy® Blood and Tissue Kit and treated with RNase A (5PRIME) following the manufacturer's protocol and stored in distilled water until GT-seq library preparation.

**Table 1. Sample size and distribution for brown (*Rattus norvegicus*) and black rats (*R. rattus*) collected across Haida Gwaii, BC.** Samples were either genotyped using double-digest restriction site-associated DNA sequencing (ddRAD) or genotyping-in-thousands by sequencing (GT-seq). All brown rats were genotyped at 315 SNPs, while all black rats were genotyped at 429 SNPs. Numbers in brackets indicate sample size less replicate samples. (*) These samples are referred to as "Kunghit" for the black rats. (†) These samples are referred to as "Lyell-SW" for the black rats. Haida island names are indicated in parentheses.

| Island | Sequencing method | Sample size | |
|---|---|---|---|
| | | **Brown rats** | **Black rats** |
| Agglomerate (K'aadxwa Xyangs Gwaayts'ads) | GT-seq | 18 | - |
| Post-Bischofs (Kingts'ii Gwaay.yaay) | ddRAD | 23 (21) | - |
| | GT-seq | 5 | |
| Pre-Bischofs | ddRAD | 31 (28) | - |
| | GT-seq | 6 (5) | |
| Ellen (Kilgii Gwaay) | ddRAD | 19 | - |
| | GT-seq | 4 | |
| Faraday (K'aaxada Gwaay) | ddRAD | 7 (6) | 14 |
| | GT-seq | 6 (5) | 4 |
| N-Graham (T'aaxwii Xaaydaɢa Gwaay.yaay iinagwaay) | ddRAD | - | 24 |
| | GT-seq | | 4 |
| S-Graham | ddRAD | - | 23 |
| | GT-seq | | 4 |
| Hotspring (Gandll K'in Gwaay.yaay) | GT-seq | 1 | - |
| Huxley (Gaaduu Gwaay) | ddRAD | - | 15 |
| | GT-seq | | 4 |
| Kunga (K'ang.ɢuu Gwaay.yaay) | ddRAD | 33 (32) | - |
| | GT-seq | 4 | |
| E-Kunghit (Gangxid Gwaay.yaay) | ddRAD | 27 | - |
| | GT-seq | 4 | |
| NW-Kunghit | ddRAD | 40 (36) | 4* |
| | GT-seq | 4 | 0 |
| Lyell (Hlɢaa Gwaay) | ddRAD | 33 (32) | 51† |
| | GT-seq | 5 | 4† |
| Lyell-SW | ddRAD | - | 59 (56) |
| | GT-seq | | 5 (4) |
| Murchison (Gaysiigas Gwaay) | ddRAD | 11 (10) | 22 |
| | GT-seq | 6 (5) | 4 |
| Prince Rupert | ddRAD | 1 | - |
| | GT-seq | 1 | |
| Ramsay (Xiina Gwaay.yaay) | GT-seq | 1 | - |
| Richardson (Sɢaanagwaay Gwaay.yaay) | ddRAD | 30 | - |
| | GT-seq | 4 | |
| Sandspit (K'il Kun Llnagaay) | ddRAD | - | 5 |
| | GT-seq | | 4 |
| Shuttle (Gwaay Daaɢaaw) | ddRAD | - | 24 |
| | GT-seq | | 4 |
| Tanu (T'aanuu Gwaay) | ddRAD | 26 | - |
| | GT-seq | 4 | |
| Tlell (Tll.aal) | ddRAD | 14 (13) | - |
| | GT-seq | 5 | |
| Total | ddRAD | 295 (281) | 241 (238) |
| | GT-seq | 78 (75) | 37 (36) |

## SNP discovery, quality control, and panel selection

Raw sequencing reads were first demultiplexed from the ddRAD samples using *process_rad-tags* as implemented in STACKS v2.0b [41]. Reads were trimmed to 100 bp to remove low-quality base calls at the 3' ends. Individual reads were aligned to the most-current brown rat reference genome (Rnor_6.0, GenBank assembly accession: GCA_000001895.4) using the default settings in Bowtie 2 v2.2.9 [42]. Single nucleotide polymorphisms were identified and genotyped using *gstacks* and *populations* in STACKS v2.0b. To ensure only high-quality SNPs were retained, all loci were filtered such that they were genotyped in >90% of total individuals, had a minimum minor allele frequency of 5%, and had a maximum observed heterozygosity of 50%. Additionally, only a single SNP per RADtag was retained to reduce the likelihood of linkage disequilibrium among loci.

To ensure maximum quality and functionality of the panel, we additionally filtered loci such that they were variable in at least one species, genotyped in >90% of individuals in each species separately, as well as with a mean depth of coverage of 6x in each species separately using VCFtools v0.1.15 [43]. Of these, only SNPs found within 40–60 bp of the start of the RADtag sequence were retained to ensure sufficient flanking sequence remained for primer design. We then estimated population differentiation for each locus both within and between species using Weir and Cockerham's unbiased estimator θ [44]. Estimates were calculated for all pairwise observations as implemented in Genetix v4.05.2 [45]. To ensure linkage equilibrium, locus pairs were positioned at least 1 Mb apart within the genome; in the event of a conflict, the locus with the lowest θ value was removed. We retained the top 350 loci with the highest discriminatory power (*i.e.*, highest θ) within each species as well as an additional eight loci for differentiating between species for a total putative panel of 708 SNPs. The associated RADtag sequences for these loci were sent to GTseek LLC (https://gtseek.com/), where custom PCR1 primers were designed for each locus incorporating Illumina primer binding sites.

## Genotyping-in-thousands by sequencing and panel optimization

To test the efficacy of our GT-seq panel, we sequenced and genotyped 92 individuals across both species (brown rats, *n* = 59; black rats, *n* = 37), preferentially selecting individuals with the highest quality ddRAD data and incorporating the entire geographic sampling distribution (Table 1). Of these, four individuals were replicated to explicitly assess genotyping error rate (see below), resulting in a total 96 samples in our initial library. GT-seq library preparation followed the original protocol [24] with some modifications [46]. DNA extractions for all individuals were normalized to 20 ng/uL to ensure even amplification across samples and diluted the PCR1 products 1:10 before use in PCR2. We quantified PCR2 products using PicoGreen™ (Molecular Probes, Inc.) and manually normalized these products to a concentration of 10 ng/μL. We pooled 2.5 μL into a final sequencing library and purified this library using the MinElute PCR Purification kit (Qiagen®), eluting into a final volume of 50 μL of nuclease-free water. The library was sequenced using a single lane of Illumina MiSeq paired-end 100 bp sequencing at the McGill University and Génome Québec Innovation Centre.

We genotyped raw sequencing data using the GT-seq pipeline found on GitHub (https://github.com/GT-seq/GT-seq-Pipeline). We used the *GT-seq_SeqTest.pl* and *GT-seq_Primer-Interaction-Test.pl* scripts to identify over-represented primer sequences and PCR artifacts and primer-dimers between loci. We removed loci accounting for >1% of total forward primer reads, loci with total counts >1% of PCR artifacts, as well as one locus per hetero-dimer pair. The remaining loci were combined into a new primer pool, and the library was re-sequenced and optimized following the above protocol. Loci that did not genotype in the final library, as well as individuals with >50% missing data, were removed from downstream analysis.

## Assessment of population structure and panel validation

To assess genotyping accuracy of the panel within each species, we compared individual genotypes from GT-seq to ddRAD at species-specific loci and measured the percent discordance across methods. We likewise calculated genotyping error rates within each sequencing method by comparing individual genotypes between replicate samples, also measured as percent discordance. Loci with missing data in either one or both of each pair of samples were not included in error calculations.

We examined the ability of our panel to differentiate between species across all panel loci using discriminant analysis of principle components (DAPC) as implemented in the R package *adegenet v*2.1.1 [47]. This procedure first applies principle component analysis (PCA) to identify genetic clusters and then uses discriminant analysis to maximize the variation among these clusters while minimizing within-cluster variation [48]. These results were compared to species-assignments from Sjodin *et al.* [25] to assess the accuracy of assignment. To ensure our subset of loci resulted in similar patterns of population structure, we ran projected PCA using the ddRAD samples to infer eigenvectors and compared to Sjodin *et al.* [25] for accuracy using the *smartpca* function in EIGENSOFT v7.2.1 [49–51]. We then projected GT-seq samples onto these eigenvectors to compare clustering across methods. Additionally, we estimated admixture coefficients for ddRAD using sparse non-negative matrix factorization (SNMF) as implemented by the *snmf()* function in the R-package *LEA v*2.6.0 [52]. We averaged admixture coefficients over 20 iterations at the previously identified optimal $k = 9$ for each species and compared these results to Sjodin *et al.* [25]. We also conducted population assignment following the method of Rannala and Mountain [53] as implemented in Geneclass2.0 [54]. Reference populations were defined using only the ddRAD samples, and both ddRAD and GT-seq samples were posteriorly assigned to these populations.

## Assignment of novel invasions

We evaluated the utility of the panel for assigning individuals of unknown origin to known source populations. We examined samples collected from novel invasions on Agglomerate Island ($n = 18$), Ramsay Island ($n = 1$), and Hotspring Island ($n = 1$) genotyped using GT-seq. To identify the source of these invasions, we conducted projected PCA using the *smartpca* function from EIGENSOFT v7.2.1 [49–51] and the ddRAD data to define eigenvectors. As the brown rats in Haida Gwaii are strongly clustered to northern, central, and southern regional populations [25], we ran the projected PCA using only the regional population to which the unknown samples most closely clustered during initial evaluation of population structure (see above). We also assigned individuals to reference populations using the method outlined in Rannala and Mountain [53] as implemented in Geneclass2.0 [54], including all potential reference populations. For this analysis, the Faraday and Murchison Island populations were grouped into a single reference population based on previous results [25].

## Results

### GT-seq panel quality and species determination

From the initial 708 putative loci, we were able to design primers targeting 526 loci. Following optimization, we retained 443 loci across both species, including 3 of 8 SNPs that were diagnostically distinct between brown and black rats. Overall, this panel had high discriminatory power to the species-level, resulting in 100% concordance with ddRAD species assignment results based on 27,686 SNPs. In the optimized GT-seq panel, all 443 loci were significantly differentiated between brown and black rats, with 150 loci exhibiting pairwise $\theta > 0.90$. From

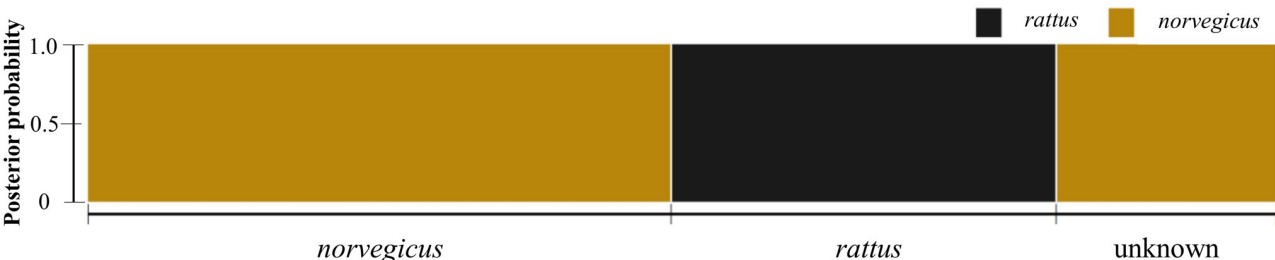

**Fig 2. Posterior probability of species assignment for brown (*Rattus norvegicus*; *n* = 53; brown) and black rats (*R. rattus*; *n* = 35; black) collected in Haida Gwaii, BC.** Rats were genotyped using a GT-seq panel of 443 SNPs. Posterior probabilities were calculated by discriminant analysis of principle components as implemented in the R-package *adegenet v*2.1.1 [47]. "Unknown" individuals were from novel invasions on Agglomerate Island (*n* = 18), Ramsay Island (*n* = 1) and Hotspring Island (*n* = 1) and were not previously assigned to species.

this optimized set of 443 loci, 315 and 429 were variable within brown and black rats, respectively, and used as species-specific datasets.

Mean GT-seq genotyping rates for brown rats was 88.6%; five individuals were removed due to a genotyping rate of <50%. Mean GT-seq genotyping rates for black rats was 78.2%; two individuals were removed due to a genotyping rate of <50%. Mean genotyping discordance across methods (ddRAD v. GT-seq) was 2.8% and 3.5% in brown and black rats, respectively. No genotyping error was observed between GT-seq replicates in brown rats; black rats revealed only a 0.2% genotyping error between GT-seq replicates. All GT-seq individuals accurately assigned to species, and all individuals from the novel invasions on Agglomerate, Hotspring, and Ramsay Islands were confirmed as brown rats (Fig 2).

## Assignment accuracy of brown rats

We detected the same three regional clusters in the brown rats as previously identified [25] via PCA (Fig 3a). Furthermore, GT-seq samples accurately projected to their sampled ddRAD clusters. Ancestry coefficients estimated at the reduced panel were consistent with previous results, though there was a loss of some finer-scale structure relative to the previous ddRAD analysis [25], especially among centrally located islands (Fig 4). Population assignment resulted in one individual sampled on the Bischof Islands in the GT-seq dataset assigning to a population other than the one in which it was sampled (Tanu Island), though this individual was previously identified as a first-generation migrant from Richardson Island in the full ddRAD analysis [25] (S1 Table).

## Assignment accuracy of black rats

Black rats also clustered similar to previous PCA results based on ddRAD [25] (Fig 5). As with the brown rats, admixture coefficients estimated using the reduced SNP panel were consistent with previous results, with all individuals accurately assigned to their sampled population (S2 Table). However, there was a loss of some finer-scale structure among the Graham Island and Sandspit populations, as well as within the Lyell Island populations relative to the ddRAD dataset [25] (Fig 6).

## Identification of origin for novel invasions

All samples from novel invasions clustered with the central brown rat populations when projected onto eigenvectors inferred using all reference populations (Fig 3a); as such, we only considered central populations as putative sources. The projected PCA for the Agglomerate,

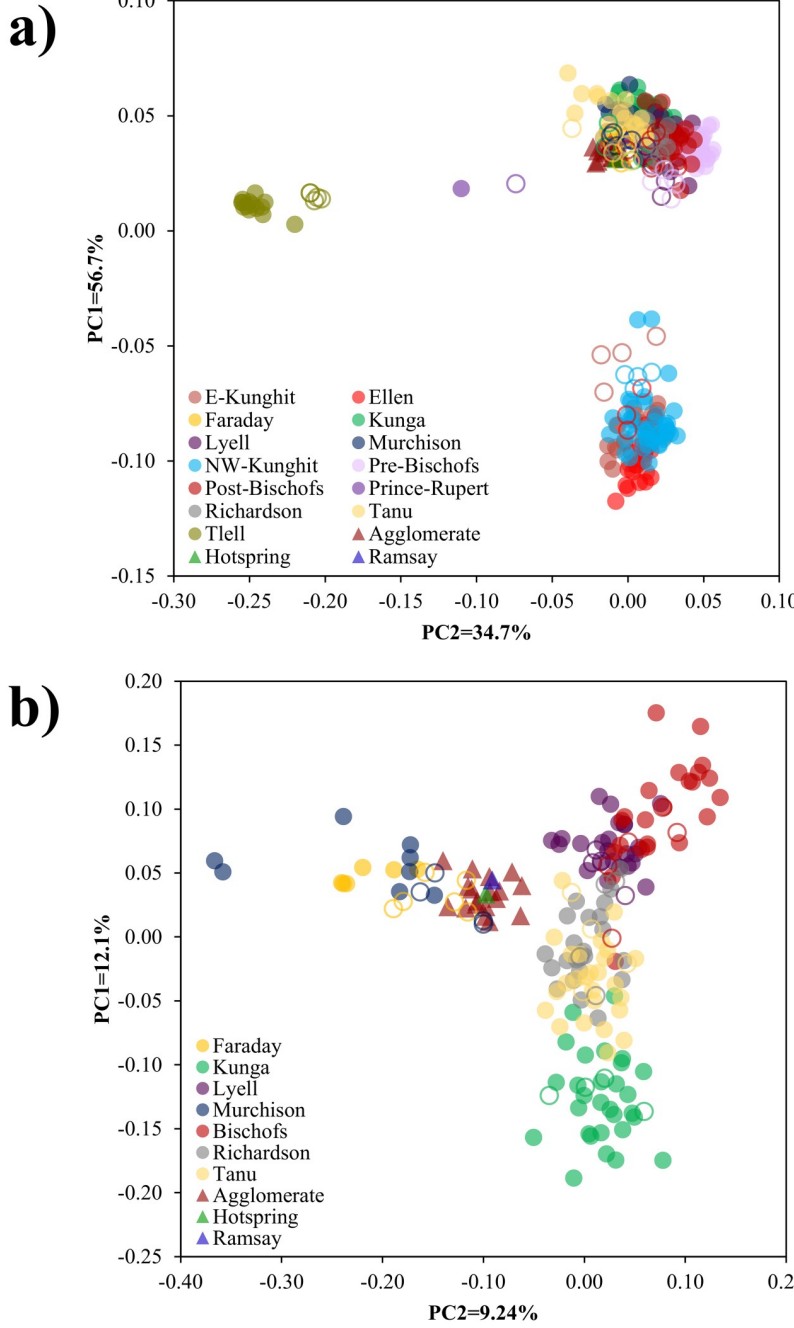

**Fig 3.** (a) Principle component analysis (PCA) for brown rats (*Rattus norvegicus*, *n* = 338) collected throughout Haida Gwaii, BC. Eigenvectors were inferred using ddRAD samples (filled circles; *n* = 265), with GT-seq samples (hollow circles, triangles; *n* = 73) posteriorly projected onto inferred eigenvectors. All analyses were performed using the same 315 loci, and analysis was implemented using EIGENSOFT v7.2.1 [49–51]. PCA was re-run using only populations centrally-located in Haida Gwaii (b) for assignment of individuals from novel island invasions (triangles, *n* = 20).

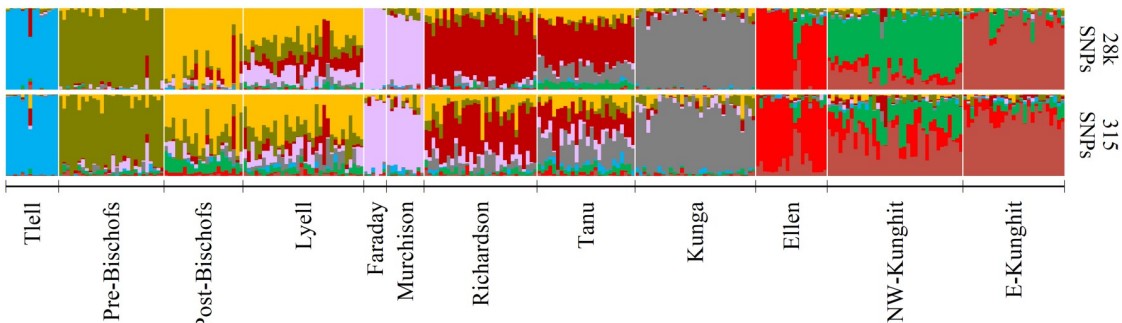

**Fig 4. Admixture coefficients for _n_ = 281 brown rats (_Rattus norvegicus_) sampled in Haida Gwaii, BC.** Admixture coefficients were calculated using the full ddRAD dataset of 27,686 SNPs (top) or with the reduce GT-seq panel of 315 SNPs (bottom). Estimates were obtained using the _snmf()_ function in the R-package _LEA v_2.6.0 [52] at an optimal _k_ = 9 and averaged over 20 iterations.

Hotspring, and Ramsay Island samples indicated a Faraday or Murchison Island origin, with these samples closely clustering with Faraday and Murchison Island projected GT-seq samples (Fig 3b). Population assignment of the Agglomerate, Hotspring, and Ramsay Island individuals strongly indicated a Faraday/Murchison Island origin (Table 2).

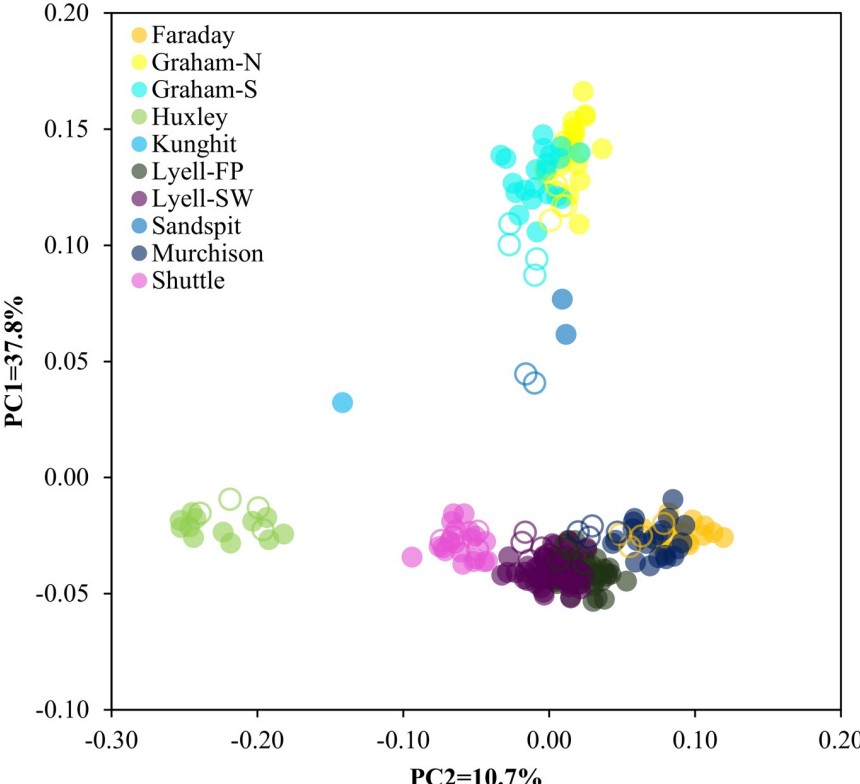

**Fig 5. Principle component analysis (PCA) for black rats (_Rattus rattus_, _n_ = 249) collected throughout Haida Gwaii, BC.** Eigenvectors were inferred using ddRAD samples (filled circles; _n_ = 214), with GT-seq samples (hollow circles; _n_ = 35) posteriorly projected onto inferred eigenvectors. All analyses were performed using the same 429 loci, and implemented using EIGENSOFT v7.2.1 [49–51].

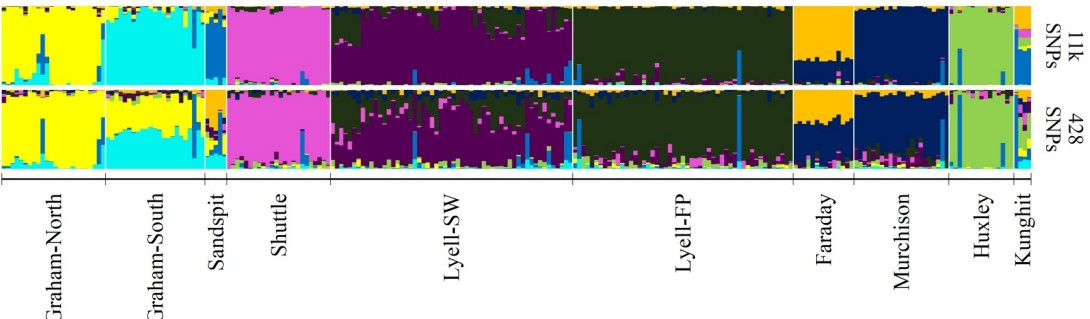

**Fig 6. Admixture coefficients for 238 black rats (*Rattus rattus*) sampled in Haida Gwaii, BC.** Admixture coefficients were calculated using the full ddRAD dataset of 10,770 SNPs (top) or with a reduce GT-seq panel of 428 SNPs (bottom). Estimates were obtained using the *snmf()* function in the R-package *LEA* *v*2.6.0 [52] at an optimal *k* = 9 and averaged over 20 iterations.

## Discussion

Invasive rats are having devastating impacts on biodiversity globally, necessitating the development of innovative tools for informing IAS management. Native species, especially seabirds, are unlikely to adapt to the invasive threat in time to prevent extirpation [55]. Without intervention, these losses could result in large scale changes at the ecosystem level [11,56–58]. Given the isolation of the Haida Gwaii archipelago, seabirds play a vital role in nutrient cycling

**Table 2. Population assignment probabilities for 20 recently invaded brown rats (*Rattus norvegicus*) of unknown ancestry sampled on three islands in Haida Gwaii, BC.** Individuals were assigned following the methods of Rannala and Mountain [53] as implemented in Geneclass2 [54]. (Far/Mur) These samples were collected from Faraday and Murchison Island and grouped into a single unit based on a previous study [25]. Only reference populations with >0 probability of assignment in at least one individual are shown.

| Sample ID | Sample Population | Probability of assignment | | | | |
|---|---|---|---|---|---|---|
| | | Far/Mur | Tanu | Richardson | Bischofs | Lyell |
| AGG-19-001 | Agglomerate | 99.99 | 0.01 | 0.00 | 0.00 | 0.00 |
| AGG-19-002 | Agglomerate | 100.00 | 0.00 | 0.00 | 0.00 | 0.00 |
| AGG-19-003 | Agglomerate | 100.00 | 0.00 | 0.00 | 0.00 | 0.00 |
| AGG-19-004 | Agglomerate | 99.56 | 0.00 | 0.44 | 0.00 | 0.00 |
| AGG-19-005 | Agglomerate | 100.00 | 0.00 | 0.00 | 0.00 | 0.00 |
| AGG-19-006 | Agglomerate | 100.00 | 0.00 | 0.00 | 0.00 | 0.00 |
| AGG-19-007 | Agglomerate | 99.70 | 0.05 | 0.22 | 0.03 | 0.00 |
| AGG-19-008 | Agglomerate | 100.00 | 0.00 | 0.00 | 0.00 | 0.00 |
| AGG-19-009 | Agglomerate | 99.94 | 0.00 | 0.06 | 0.00 | 0.00 |
| AGG-19-010 | Agglomerate | 100.00 | 0.00 | 0.00 | 0.00 | 0.00 |
| AGG-19-011 | Agglomerate | 100.00 | 0.00 | 0.00 | 0.00 | 0.00 |
| AGG-19-012 | Agglomerate | 100.00 | 0.00 | 0.00 | 0.00 | 0.00 |
| AGG-19-013 | Agglomerate | 100.00 | 0.00 | 0.00 | 0.00 | 0.00 |
| AGG-19-014 | Agglomerate | 98.60 | 1.35 | 0.04 | 0.00 | 0.00 |
| AGG-19-015 | Agglomerate | 99.98 | 0.00 | 0.00 | 0.00 | 0.02 |
| AGG-19-016 | Agglomerate | 100.00 | 0.00 | 0.00 | 0.00 | 0.00 |
| AGG-19-017 | Agglomerate | 100.00 | 0.00 | 0.00 | 0.00 | 0.00 |
| AGG-19-018 | Agglomerate | 99.99 | 0.01 | 0.00 | 0.00 | 0.01 |
| HOT_RAT | Hotspring | 100.00 | 0.00 | 0.00 | 0.00 | 0.00 |
| RAM-RAT | Ramsay | 100.00 | 0.00 | 0.00 | 0.00 | 0.00 |
| Mean | N/A | 99.89 | 0.07 | 0.04 | 0.00 | 0.00 |

as significant transporters of nutrients from off-island via guano deposition [58,59]. Disruption of this transport chain could lead to increased depletion of resources on the islands and whole-scale regime shifts, a consequence of rat invasions observed in other island systems around the globe [16,58–60]. The effects of removing seabirds from Haida Gwaii are not purely ecological; seabirds, such as the rat-threatened ancient murrelet (*Synthliboramphus antiquus*), also have a significant cultural importance to the Haida Nation [31]. To promote seabird recovery in Haida Gwaii, invasive rats must be eradicated.

Recent work has shown the utility of genetic data for informing invasive rat management in Haida Gwaii [25]. By harnessing the power of next-generation sequencing and ddRAD, previous work characterized population structure and connectivity among both brown and black rats throughout the archipelago, laying the foundation for robust inference of population origin for nearly any novel invasion [25]. To demonstrate application, this earlier study revealed that failed eradications on the Bischof Islands were due to re-invasion from nearby Lyell Island and not from incomplete removal [*i.e.*, the survivor hypothesis; 25]. The mechanism of reinvasion was likely via swimming given the close geographic proximity (<500m) to Lyell Island. "Rafting" during high tide and/or commensal spread via ships remain possibilities. Additionally, the origin of novel invasions on Faraday and Murchison Islands was also identified as neighboring Lyell Island [25]. These results not only highlighted the need for potentially increased biosecurity between islands in Haida Gwaii, but also provided insights into the dispersal capacity of brown rats through temperate waters.

This initial population genomic analysis was achieved using ddRAD, a reduced representation genome sequencing approach that allows for the genotyping of dozens or more individuals at a large number of SNPs (>10,000) in a single lane of Illumina sequencing [61]. Yet, this method requires labour-intensive library preparation, where speed and cost-effectiveness do not scale linearly for smaller sample sizes. These considerations are of the utmost importance, as novel rat invasions often occur with relatively few individuals [62,63]; in some cases, only a single individual may be captured for analysis when populations are at low densities (for example, on Hotspring and Ramsay Islands). In addition, ddRAD requires substantial bioinformatic processing and filtering of data post-sequencing to ensure genotype calls are accurate and not the result of sequencing error, sample contamination, or other confounding factors [64]. While ddRAD does afford connectivity between datasets, these bioinformatic processing steps must be repeated for inclusion of any new individuals. Taken together, ddRAD and similar approaches are not ideal when applied to a single or few samples, especially when the required information is time-sensitive. Rat invasions, in particular, have been likened to other acute impacts to biodiversity such as wildfires or oil spills that require rapid and decisive responses based on timely information [65].

GT-seq overcomes these limitations, providing a more expedient and cost-effective approach for generating needed genetic information, especially when involving a single or few high priority samples. To start, GT-seq library preparation is highly simplified compared to ddRAD [24,61]. While ddRAD has many time-consuming purification and size selection steps along with multiple ligations and PCR steps, GT-seq only requires two PCRs and a final library purification. Consequently, GT-seq libraries can be prepared in a matter of hours rather than days or weeks as may be the case with ddRAD. Library preparation for GT-seq uses few reagents and a commercially available PCR kit, whereas ddRAD library preparation requires a fuller suite of more costly reagents including restriction enzymes and ligases. Moreover, bioinformatic processing and analyses of raw data are also much more streamlined in GT-seq, which uses open-source software (https://github.com/GTseq/GTseq-Pipeline) to simultaneously identify locus-specific primers and genotype the corresponding SNP using minimal computational resources (*e.g.*, standard laptop). These genotypes require no additional

filtering and are immediately suitable for use in population genetic analyses, further positioning GT-seq as an effective rapid response tool.

While GT-seq does offer many advantages over other approaches (*e.g.*, reduced representation sequencing, including ddRAD), panel development does require existing genomic data for locus identification and primer design [24,46]. For this study, we had previously genotyped brown and black rats across their distributions in Haida Gwaii, as well as characterized population structure among island populations [25]. This previous work not only provided substantial genomic data for GT-seq panel design, but also allowed for an explicit evaluation of the ability of the panel to identify known population structure based on ddRAD results. While we were fortunate in this regard, many systems are lacking these foundational data. In such systems, prior genetic work will be needed to best inform GT-seq panel development; however, once completed, the full advantages of GT-seq over more costly and labour-intensive methods can be exploited [24,46].

Moving forward, GT-seq and other targeted amplicon sequencing approaches hold great promise for informing conservation and management of biodiversity [66]. Originally demonstrated in steelhead trout [24], GT-seq has been subsequently applied to the study of other fish species, including redband trout [66], brook trout [67], Pacific lamprey [68], and walleye [69]. More recently, the efficacy of using GT-seq for genotyping minimally-invasive samples has been demonstrated [46], and successfully applied to further understanding of the molecular ecology and conservation status of the at-risk Western rattlesnake [70]. To the best of our knowledge, our study is the first to demonstrate use of GT-seq to inform invasive species management to help mitigate biodiversity loss.

## Panel efficacy

Across both species-specific panels, we were able to accurately and consistently detect previously identified population structure [25] among invasive rat populations in Haida Gwaii. Principle component analysis resulted in near identical clustering when compared to the previous ddRAD study [25] (Figs 3 and 5). We also identified genetic structure consistent with previous results, though there was a slight loss in resolution of fine-scale structure among highly connected populations (Figs 4 and 6) likely due to the reduced number of SNPs in the GT-seq dataset. Both brown and black rats are highly vagile species; significant connectivity and gene flow can occur among island populations separated by as much as 1 km of ocean, and terrestrial populations can remain connected at distances of >10 km [38], resulting in less differentiation and more admixture. Using large SNP datasets ($\leq$ 25,686 loci), Sjodin *et al*. [25] were able to detect this fine-scale structure even among minimally differentiated populations. To detect this same structure using our reduced GT-seq panel, targeted SNPs likely will need to be added to further maximize among-cluster variation for these admixed clusters, especially for brown rats sampled on Lyell, Tanu, and Richardson Islands, and for both species with sampling distributions across larger islands like Graham and Kunghit Islands. Nevertheless, these panels still adequately captured among-population variation across both species capable of correctly assigning individuals to their true populations. Furthermore, as GT-seq is a targeted amplicon sequencing approach, this method is highly replicable and completely connectible, so additional reference populations can always be added to the database to allow for even more robust inferences. The ability to cost-effectively genotype individuals is also conducive to applications such as long-term monitoring to better estimate contemporary gene flow and inform biosecurity measures. These estimates would be particularly useful for monitoring brown rat populations on Lyell and surrounding islands, as these populations appear to be rapidly expanding and are currently the highest biosecurity threat in terms of seabird conservation [25,31].

## Novel invasions

We found strong support for a Faraday and/or Murchison origin for novel invasions on Agglomerate, Hotspring, and Ramsay Islands (Fig 3b, Table 2). The Ramsay Island invasion is particularly concerning as this island hosts the most significant remaining seabird colony in the entire archipelago [31]. Rat invasions have already led to the complete, or near complete, extirpation of large seabird colonies in Haida Gwaii, such as on Langara, Kunghit and Lyell Islands [37,71]; establishment of a rat population on Ramsay Island will likely lead to the same devasting consequences for extant seabird populations. Prior to this study, we investigated the origin of the incursion onto Hotspring Island using ddRAD (Sjodin & Russello, unpublished report). These results also indicated a Faraday/Murchison Island origin and provided information to support an eradication on Hotspring Island in November 2018 with the goal of preventing spread to Ramsay Island. But for reasons discussed above, ddRAD, while accurate, is not suitable for a single or few individuals as resources and processing time do not scale down linearly for small sample sizes. For island eradications and control of incursions, rapid response is critical and targeting the response at an appropriate scale is crucial to the durability of the eradication. Eradicating the rats on Hotspring Island kept Ramsay Island rat-free, but only for a year; rats subsequently re-invaded Hotspring Island, which likely acted as a stepping-stone leading to recent detections on Ramsay Island (Wojtaszek, pers. comm.). This outcome indicates that the management response to the initial incursions on Hotspring Island was conducted at an inappropriate geographic scale; the distance from Ramsay Island that needed to remain rat free to provide some eradication durability was greater than just the most adjacent islands (Wojtaszek, pers. comm.). Armed with information from the RapidRat GT-seq panel, future initiatives could determine whether all islands within a certain area should be considered a single eradication unit or if a more localized response is appropriate. These data can also point to mechanisms of invasion, which can guide how management agencies deal with the incursion. For example, if the rat on Hotspring Island had come from another location within the archipelago via a vessel moored on the north side of the island (*e. g*., human-mediated), rather than via natural dispersal from adjacent islands, the response would be scaled, and biosecurity measures enhanced.

## Conclusions

Here, we developed, validated and deployed an effective and efficient genotyping tool (RapidRat) for informing invasive rat management in Haida Gwaii. It is important to highlight, however, that GT-seq panel development was greatly facilitated by the existence of an archipelago-wide SNP dataset previously collected via ddRAD [25]. When no such data are available, an initial population genomic analysis would most likely need to be conducted. Moreover, we initially designed RapidRat to be informative for both invasive brown and black rats, likely limiting the resolution at the single species level. Although, this does not appear to be an issue for informing invasive rat management in Haida Gwaii, independent, species-specific panels could be developed using a similar workflow as described here. The potential multi-species ascertainment bias also likely translates to a finer-level; for example, we do not expect RapidRat to be informative in other systems given that locus selection was solely based on among-population genetic variation in Haida Gwaii. Nevertheless, invasive rats constitute a global threat to biodiversity and innovative tools are required to inform management action and minimize impact on novel ecosystems. RapidRat demonstrates the applicability of GT-seq as one such IAS rapid response tool. Importantly, the framework demonstrated here for panel development and validation can be directly applied for informing invasive rat management and biodiversity conservation in other systems.

## Supporting information

**S1 Table. Population assignment probabilites for n = 53 brown rats (*Rattus norvegicus*) sampled in Haida Gwaii, BC.** Individuals were assigned to ddRAD reference populations using the methods outlined in Rannala and Mountain (1997) as implemented in the program Geneclass2 (Piry et al. 2004). Highlighted lines indicate samples that assigned to population other than their sampled population.
(XLSX)

**S2 Table. Population assignment probabilites for n = 34 black rats (*Rattus rattus*) sampled in Haida Gwaii, BC.** Individuals were assigned to ddRAD reference populations using the methods outlined in Rannala and Mountain (1997) as implemented in the program Geneclass2 (Piry et al. 2004).
(XLSX)

## Acknowledgments

We thank Parks Canada Agency employees from Gwaii Haanas National Park Reserve, National Marine Conservation Area Reserve and Haida Heritage site for the collection of the field samples. We are particularly grateful to Danielle Schmidt and Nathan Campbell for providing support with GT-seq panel development and optimization.

## Author Contributions

**Conceptualization:** Michael A. Russello.

**Data curation:** Bryson M. F. Sjodin, Robyn L. Irvine.

**Formal analysis:** Bryson M. F. Sjodin.

**Funding acquisition:** Michael A. Russello.

**Project administration:** Michael A. Russello.

**Supervision:** Michael A. Russello.

**Writing – original draft:** Bryson M. F. Sjodin.

**Writing – review & editing:** Robyn L. Irvine, Michael A. Russello.

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
