## [Decision Letter · Decision Letter 0]

1 Apr 2020

PONE-D-20-00793

RapidRat: development, validation and application of a genotyping-by-sequencing panel for rapid biosecurity and invasive species management

PLOS ONE

Dear Dr. Russello,

Thank you for submitting your manuscript to PLOS ONE. After careful consideration, we feel that it has merit but does not fully meet PLOS ONE’s publication criteria as it currently stands. Therefore, we invite you to submit a revised version of the manuscript that addresses the points raised during the review process.

We would appreciate receiving your revised manuscript by June 30th 2020. To enhance the reproducibility of your results, we recommend that if applicable you deposit your laboratory protocols in protocols.io, where a protocol can be assigned its own identifier (DOI) such that it can be cited independently in the future. For instructions see: http://journals.plos.org/plosone/s/submission-guidelines#loc-laboratory-protocols

We look forward to receiving your revised manuscript.

Kind regards,

Daniel de Paiva Silva, Ph.D.

Academic Editor

PLOS ONE

Journal Requirements:

1. Thank you for including the following ethics statement:

'All sample collection was performed under Parks Canada Agency Animal Care Committee protocol GHNPR11-5. Parks Canada is the regulatory agency that permits

field research in the study area and were the sole participants involved in sample collection.'

Please specify the ethics statement in the Methods section of your manuscript and ensure that your named ethics committee specifically approved this study.

For additional information about PLOS ONE ethical requirements for animal research, please refer to http://journals.plos.org/plosone/s/submission-guidelines#loc-animal-research

2. Thank you for including your competing interests statement;"The authors have declared that no competing interests exist."

We note that author Michael Russello is an Academic editor for PLOS ONE

4.

Additional Editor Comments (if provided):

Dear Mr. Russello,

after carefull reviews by two independent researchers, they found that your manuscript is nearly to be accepted. Nonetheless, they suggested that minor reviews are performed before it is formally accepted in the journal, especially reviewer #2, who provided some suggetions in an attached file. Therefore, considering the whole COVID-19 pandemic, I hope you are able to provide a reviewed version of the manuscript by June 30th 2020.

Best regards,

Daniel Silva

Reviewers' comments:

Reviewer's Responses to Questions

**Comments to the Author**

1. Is the manuscript technically sound, and do the data support the conclusions?

Reviewer #1: Yes

Reviewer #2: Yes

2. Has the statistical analysis been performed appropriately and rigorously? 

Reviewer #1: Yes

Reviewer #2: Yes

3. Have the authors made all data underlying the findings in their manuscript fully available?

Reviewer #1: Yes

Reviewer #2: Yes

4. Is the manuscript presented in an intelligible fashion and written in standard English?

Reviewer #1: Yes

Reviewer #2: Yes

5. Review Comments to the Author

Reviewer #1: The paper presents novelty and describes the way the authors obtained genomic data to evalute invasive species. The authors developed, validated and deployed an effective and efficient genotyping tool (RapidRat) for informing invasive rat management in Haida Gwaii.

To improve the paper I recomend:

a) Make the results clear in the abstract section

b) Introduction and material and methods: are Ok.

c) Results: ok

d) Discussion and conclusions: Improve them to clarify the novelty, importance and the limitations of such research

Reviewer #2: The manuscript entitled “RapidRat: development, validation and application of a genotyping-by-sequencing panel for rapid biosecurity and invasive species management” describes the development of a rat specific panel for genomic data to access information about invasive species of rats in an archipelago in British Columbia, Canada. The authors proposed a novel and cheaper way to genotype individuals of brown and black rats using the detection of SNPs polymorphisms to track novel and old rat invasions in the islands. I recommend this manuscript to be accepted for publication in PlosOne after the review of the suggestions pointed out in the attached file.

6. PLOS authors have the option to publish the peer review history of their article (what does this mean?). If published, this will include your full peer review and any attached files.

Reviewer #1: No

Reviewer #2: No

---

## [Author Response · Author response to Decision Letter 0]

1 May 2020

Please see attached response letter

---

## [Decision Letter · Decision Letter 1]

2 Jun 2020

RapidRat: development, validation and application of a genotyping-by-sequencing panel for rapid biosecurity and invasive species management

PONE-D-20-00793R1

Dear Dr. Russello,

We are pleased to inform you that your manuscript has been judged scientifically suitable for publication and will be formally accepted for publication once it complies with all outstanding technical requirements.

With kind regards,

Daniel de Paiva Silva, Ph.D.

Academic Editor

PLOS ONE

Additional Editor Comments (optional):

Reviewers' comments:

Reviewer's Responses to Questions

**Comments to the Author**

1. If the authors have adequately addressed your comments raised in a previous round of review and you feel that this manuscript is now acceptable for publication, you may indicate that here to bypass the “Comments to the Author” section, enter your conflict of interest statement in the “Confidential to Editor” section, and submit your "Accept" recommendation.

Reviewer #2: All comments have been addressed

Reviewer #3: All comments have been addressed

2. Is the manuscript technically sound, and do the data support the conclusions?

Reviewer #2: Yes

Reviewer #3: Yes

3. Has the statistical analysis been performed appropriately and rigorously? 

Reviewer #2: Yes

Reviewer #3: Yes

4. Have the authors made all data underlying the findings in their manuscript fully available?

Reviewer #2: Yes

Reviewer #3: No

5. Is the manuscript presented in an intelligible fashion and written in standard English?

Reviewer #2: Yes

Reviewer #3: Yes

6. Review Comments to the Author

Reviewer #2: (No Response)

Reviewer #3: This article deals with the development of a panel of SNPs in a genotyping-by-sequencing strategy (GT-seq) and its possible application in the management of invasive species of rats in Canada. A review round has already been carried out. Although the authors have made a substantial improvement in the manuscript, I recommend some minor adjustments before being accepted for publication in PlosOne.

Introduction: ok

Material and Methods:

1) Information about the ddRAD-seq sequencing data is missing. Which sequencing platform was used to generate this data? How many RAD-tags were generated? What type of sequencing read was generated? Is this data deposited in a database? What is the mean read length? The RAD-tags represent which portion of the total size of the genome used as a reference?

2) I believe that "Study design" is not a good topic name for methods (line 131). Something like "origin of SNPs" or "SNP loci information from ddRAD-Seq" is perhaps more appropriate. After all, the other topics of the methodology are also part of the "study design".

Results: Ok

Discussion: Ok

Conclusion: Ok

Figures and tables:

As mentioned by reviewer # 2 in the previous review, the figures are still in low resolution.

7. PLOS authors have the option to publish the peer review history of their article (what does this mean?). If published, this will include your full peer review and any attached files.

Reviewer #2: No

Reviewer #3: Yes: Rhewter Nunes

---

## [Editor Report · Acceptance letter]

10 Jun 2020

PONE-D-20-00793R1 

RapidRat: development, validation and application of a genotyping-by-sequencing panel for rapid biosecurity and invasive species management 

Dear Dr. Russello:

I'm pleased to inform you that your manuscript has been deemed suitable for publication in PLOS ONE. Congratulations! Your manuscript is now with our production department. 

Kind regards, 

on behalf of

Dr. Daniel de Paiva Silva 

Academic Editor

PLOS ONE